# FROM SPATIAL TRANSCRIPTOMICS TO TOKENS: GENERATIVE PRE-TRAINING WITH BYTE-PAIR ENCODING

## ABSTRACT

Generative pre-trained models have attained extraordinary success in natural language processing and computer vision. Meanwhile, spatial single-cell transcriptomics has become a potent tool for exploring disease mechanisms. Current methods largely disregard the influence of RNA spatial arrangement on cellular identity and disease progression. This can result in the loss of RNA co-localization information, incomplete spatial transcriptome analysis, and inadequate exploration of disease mechanisms, thus failing to identify crucial approaches for clinical diagnosis. To tackle these problems, we put forward STBPE (Spatial Transcriptomics Byte Pair Encoding), a pre-training framework concentrating on subcellular resolution. This framework innovatively incorporates "spatially aware byte pair encoding strategies". It converts the subcellular localization information of RNA within a single cell into serialized token units, realizing accurate digital depiction of RNA spatial distribution patterns. Specifically, it first employs a spatial omics data-driven word segmentation algorithm to encode the spatial coordinates and transcript features of RNA into a unified byte pair sequence. Then, it utilizes the BERT-style masked self-supervised learning paradigm to randomly mask partial spatially aware labels and reconstruct the original sequence, compelling the model to learn deep embedding representations that contain spatial position information. This design allows STBPE to seize the potential correlation between RNA spatial distribution and gene expression, notably enhance cell type annotation, discover co-localized RNAs related to cellular identity from a novel viewpoint, and lay the groundwork for constructing multimodal foundation models that integrate spatial transcriptomics with natural language.

## 1 INTRODUCTION

Spatially resolved single-cell transcriptomics has emerged as a cutting-edge technology in the life sciences, gradually demonstrating its tremendous potential for elucidating disease mechanisms Chen et al. (2020), Ståhl et al. (2016). This technique not only enables the simultaneous acquisition of gene expression profiles at single-cell resolution but also preserves the precise spatial localization of transcripts within the tissue microenvironment. As a result, it provides an unprecedented perspective for understanding cellular identity, interactions, and their dynamic changes during disease onset and progression Baccin et al. (2020).

Nevertheless, existing analytical approaches remain limited. Conventional methods are largely confined to gene expression matrices and fail to effectively incorporate spatial information Wolf et al. (2018). Although a number of algorithms have recently been developed to integrate spatial context —for example, graph neural network–based spatial modeling methods Hu et al. (2021), Xu et al. (2022), Long et al. (2023), Pham et al. (2023)-which show promising potential in the analysis of cell-level spatial transcriptomics data, they still fall short of addressing the analytical demands posed by the new generation of subcellular-resolution technologies, such as 10x Xenium Janesick et al. (2023), NanoString CosMx He et al. (2022), and Vizgen MERSCOPE Chen et al. (2015).

Meanwhile, generative pre-trained models have achieved revolutionary success in natural language processing and computer vision Wang et al. (2023). Their core strength lies in leveraging large-scale self-supervised learning to capture complex patterns and contextual dependencies, thereby attaining strong generalization performance across downstream tasks Du et al. (2023). More impor-

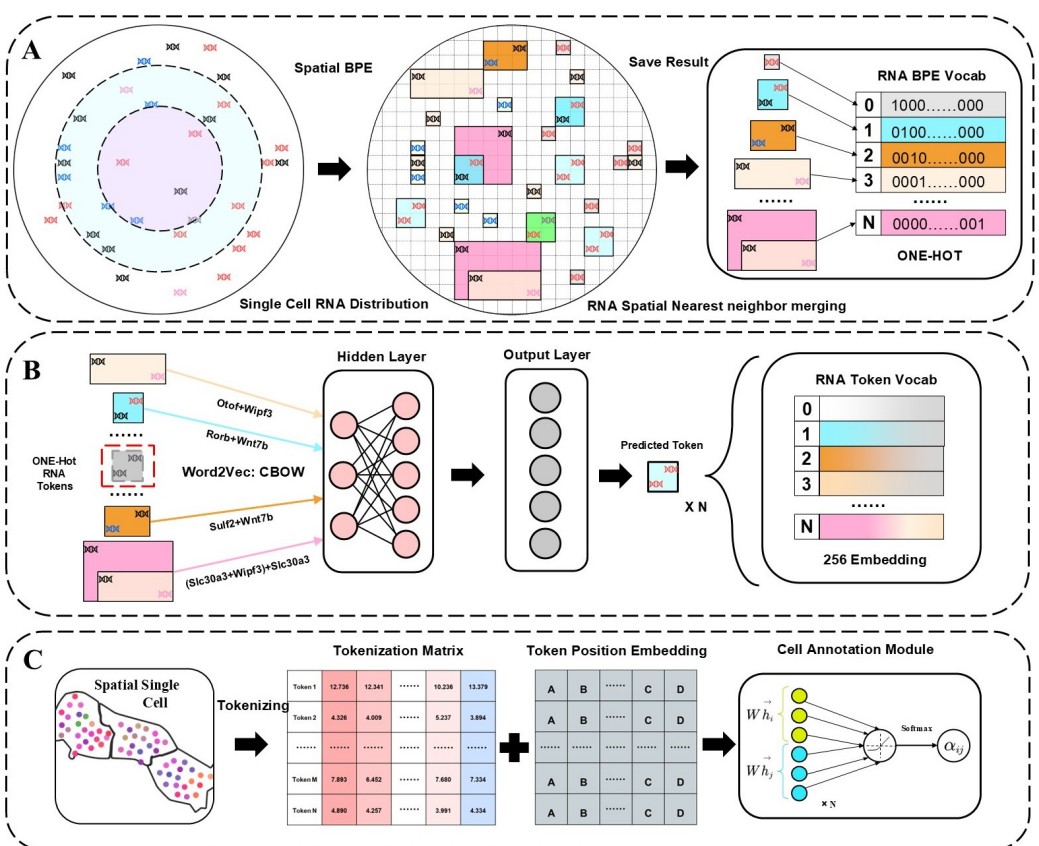

Figure 1: Overflow of STBPE

tantly, the tokenization strategies underpinning these models—such as Byte-Pair Encoding (BPE) Sennrich et al. (2015)—enable the effective transformation of high-dimensional and complex data into discrete token representations, providing a unified semantic space for model understanding and reconstruction.

In the field of single-cell research, several recent efforts have explored the integration of pre-trained models with single-cell data, demonstrating the potential for cross-modal and cross-scale modeling Wang et al. (2025). For instance, scGPT-spatial extends a single-cell foundation model to spatial transcriptomics data through continual pre-training, validating the applicability of the pre-training paradigm at the spatial scale Wang et al. (2025). scBERT adopts a BERT-style pre-training framework to model single-cell transcriptomes, thereby improving the accuracy of cell type annotation Yang et al. (2022). scGPT, inspired by generative large models, exhibits strong transfer learning and generalization capabilities across diverse single-cell omics tasks Cui et al. (2024). Moreover, Chen and Zhou explores the direct use of ChatGPT's embedding capabilities for modeling single-cell data, further underscoring the potential of natural language pre-training paradigms in single-cell biology Chen and Zou (2025). Collectively, these studies provide important insights into bridging spatial single-cell transcriptomics with generative pre-trained models.

Inspired by these advances, we introduce the generative pre-training framework into spatial single-cell transcriptomics, leveraging encoding strategies such as BPE to transform subcellular-resolution RNA spatial coordinates and expression features into serialized spatial tokens. This approach not only maximizes the preservation of RNA spatial organization but also opens up the possibility of constructing self-supervised learning models with spatial understanding capabilities. Such a cross-disciplinary integration holds the potential to overcome the limitations of existing methods, offering a novel perspective for uncovering RNA co-localization relationships and their roles in regulating cellular identity and disease mechanisms.

We propose a novel pre-training framework—STBPE (Spatial Transcriptomics Byte Pair Encoding). As illustrated in Figure 1, the core idea of STBPE is to serialize subcellular-resolution RNA spatial distributions into spatial token units and to learn their deep embedding representations within a generative pre-training model.

Specifically, STBPE first employs a Spatial BPE algorithm to segment and merge RNA spatial positions within a single cell, encoding the original point-like transcript distribution into a sparse yet structured token sequence (Fig. 1A). This process preserves the relative positional relationships among RNA molecules, such as distances and angles, thereby distinguishing it from graph-based approaches that rely solely on topology or adjacency. Next, STBPE adopts training paradigms akin to Word2Vec/CBOW and BERT-style masked learning to learn contextualized representations of RNA tokens (Fig. 1B), while constructing a unified RNA token vocabulary and embedding space. Finally, in downstream applications, STBPE incorporates positional encoding and annotation modules (Fig. 1C) to support cell type annotation, spatial pattern discovery, and the interpretation of potential biological mechanisms. Our main contributions are as follows:

- A novel spatial BPE algorithm. Inspired by natural language processing and graph-based BPE, we propose a new spatial BPE method tailored to sparse RNA distributions. Unlike traditional graph BPE, which only considers node adjacency, STBPE preserves explicit distances, angles, and relative positional information between RNA molecules during encoding, making it more suitable for modeling subcellular-resolution spatial transcriptomics data.
- The STBPE pre-training model. We construct and train the STBPE model on large-scale spatial single-cell transcriptomics data (approximately one million cells). Experiments demonstrate that our model significantly improves the accuracy of cell type annotation and exhibits superior robustness and generalization in cross-dataset transfer scenarios.
- New biological insights. By analyzing the learned token representations and relationships from STBPE, we uncover several RNA co-localization patterns and spatial regulatory signals closely associated with cellular identity. These findings not only provide new evidence for cell type discrimination but also offer fresh perspectives for investigating disease-related spatial transcriptomic abnormalities.

## 2 RELATED WORK

In spatial and single-cell transcriptomics research, cell type annotation has long been a central task. In recent years, a variety of deep learning and representation learning approaches have emerged, significantly advancing both the accuracy and efficiency of annotation. For example, DSCT proposed a novel deep learning framework that enables rapid and accurate annotation of spatial transcriptomics cell types Xu et al. (2025). scDOT enhances the robustness of single-cell RNA-seq annotation through a multi-reference integration strategy and is capable of discovering potential new cell populations Xiong and Zhang (2024). At the subcellular scale, TopACT attempts to classify spatial transcriptomics data within cells by leveraging multi-scale topological features Benjamin et al. (2024). Meanwhile, structured model designs have also demonstrated unique advantages: Tosica incorporates Transformers to provide an interpretable, all-in-one solution for cell type annotation Chen et al. (2023), while TripletCell improves clustering and annotation accuracy by optimizing the embedding space through a triplet metric learning framework Liu et al. (2023). Despite these advances, when faced with the complexity of RNA spatial distributions at subcellular resolution, challenges in generalization and fine-grained modeling still remain.

Multimodal pre-trained models have shown remarkable foresight and potential in artificial intelligence, with their core strength lying in unifying representations across different modalities and capturing latent cross-modal associations through large-scale self-supervised learning. A key prerequisite for achieving joint modeling of complex modalities within a single framework is the transformation of heterogeneous data into a unified token representation, enabling the model to learn underlying patterns via sequence modeling. Byte-Pair Encoding (BPE) has proven to be an effective tool for this purpose and has already been extended to non-linguistic domains such as vision and biology. For example, From Pixels to Tokens transforms quantized image pixels into tokens, providing new directions for visual generation Zhang et al. (2024); Multidimensional BPE reduces the sequence length of multi-dimensional data, thereby improving generative efficiency Elsner et al.

(2024); Unified Multimodal Understanding via Byte-Pair Visual Encoding explores the unification of vision and other modalities within a shared token space through BPE Zhang et al. (2025); and GraphBPE pioneers the transformation of molecular graphs into token sequences, enabling molecular generation tasks to benefit from language model paradigms Shen and Póczos (2024). In genomics, DNABERT leverages the BERT pre-training framework to effectively capture contextual features of DNA sequences Ji et al. (2021), DNAGPT introduces generative pre-training into genomic data Zhang et al. (2023), and GROVER demonstrates the ability to learn sequence contexts at scale on the human genome Sanabria et al. (2024). Collectively, these studies highlight that representing complex modalities as tokens and modeling them under a unified pre-training paradigm not only enhances representational power but also opens new avenues for spatial single-cell transcriptomics.

## 3  PROBLEM DEFINITION

Subcellular resolution spatial transcriptomics data can be formalized as a point set

$$\mathcal{C} = \{(g_i, x_i, y_i) \mid i = 1, 2, \ldots, N\},$$

where $g_i$ is the gene type of the RNA, $(x_i, y_i)$ are the 2D coordinate positions, and $N$ is the total number of RNAs in the cell. This type of data exhibits sparsity, spatial dependence, and multi-scale colocalization characteristics. Traditional BPE only merges high-frequency symbol pairs in 1D symbol sequences and cannot be directly applied to 2D format RNA data; while GraphBPE introduces adjacency structures, it often loses continuous geometric information (such as Euclidean distance and angle) at the subcellular level, making it difficult to characterize complex colocalization patterns. Therefore, we aim to represent cells as a series of **spatial tokens**, i.e., learning the mapping

$$f : \mathcal{C} \longrightarrow \mathcal{T} = \{t_1, t_2, \ldots, t_M\}, \quad M \ll N,$$

where each token $t_j$ is obtained by aggregating the subset of RNAs it covers $\Omega_j$:

$$t_j = \phi\left(\{(g_i, x_i, y_i) \mid i \in \Omega_j\}\right),$$

while explicitly preserving the distance between RNAs

$$d_{ij} = \|(x_i, y_i) - (x_j, y_j)\|,$$

and angle

$$\theta_{ij} = \arctan \frac{y_j - y_i}{x_j - x_i}.$$

The resulting token sequence not only compresses the sparse RNA distribution but also embeds spatial and colocalization relationships in the serialized representation, providing more biologically meaningful input features for pre-trained models.

## 4  METHOD

We propose a novel pre-training framework, STBPE (Spatial Transcriptomics Byte Pair Encoding), which transforms subcellular-resolution RNA spatial distributions into structured token sequences and learns their representations in a unified embedding space. The key innovation lies in the first component, where we design a new Spatial BPE method specifically tailored for sparse point distributions. This method segments and merges RNA spots within a single cell to generate sparse yet structured spatial token sequences, effectively compressing the original RNA point distribution while preserving relative spatial relationships such as distances and angles—an advantage over conventional graph-based approaches that rely solely on topology or adjacency. Building upon these token sequences, STBPE adopts training paradigms inspired by Word2Vec/CBOW and BERT-style masked modeling to learn contextual embeddings of RNA tokens and construct a unified vocabulary and embedding space. Finally, the resulting token embeddings can be applied to a range of downstream analyses, including cell type annotation, spatial pattern discovery, and the elucidation of potential biological mechanisms, offering a new methodological perspective for spatial transcriptomics in complex tissue environments.

## 4.1 STBPE Detailed Algorithm

The STBPE (Spatial Transcriptomics Byte Pair Encoding) mainly consisting of three core steps: data preprocessing, statistical high-frequency RNA pair merging, and multiple rounds of iteration. The algorithm takes single-cell RNA expression data as input, identifies spatially adjacent RNA pairs through our custom position encoding and direction judgment mechanism, and iteratively merges high-frequency combinations based on statistical frequency selection. The overall process design follows a cyclic optimization mode of "processing merging updating", gradually optimizing the spatial distribution of RNA through a preset number of iterations.For the detailed implementation of the pseudocode, please refer to Appendix D.

### 4.1.1 Data Preprocessing

To address non-uniform RNA coordinates across cells, we built a preprocessing and grid-quantization framework that maps intracellular RNA molecules and boundary information to a standardized grid. Using geometric normalization (e.g., the minimum circumcircle), cells of varying sizes and positions are aligned to a common coordinate system. A double-layer grid with a greedy, collision-free assignment ensures each RNA is uniquely placed in a fine grid, while boundary points use nearest-center alignment to preserve structure. The system supports automated batch processing across multiple cells and layers, organizes outputs via a hierarchical directory scheme, and maintains a global gene dictionary for unique identifiers. A parameter self-optimization mechanism selects grid settings based on data characteristics, balancing computational efficiency and spatial resolution.

### 4.1.2 Statistical analysis of high-frequency RNA for merging

Because RNA itself has spatial propertiessee Figure 2, when storing RNA pairs, we not only need to record the distance but also the orientation. To solve this problem, we defined 8 positional relationships (such as top left, top right, etc.) based on the relative position encoding of RNA in spatial coordinates, and provided direction reversal function to support symmetric positional relationship processing. Position encoding adopts formatted binary representation as shown in the Table 1

| Relative position | Position code |
|---|---|
| Right | 000 |
| Upper right | 001 |
| Above | 010 |
| Upper left | 011 |
| Left | 100 |
| Lower left | 101 |
| Below | 110 |
| Lower Right | 111 |

Table 1: Corresponding table of relative position and position code

At the same time, in order to reflect the specific direction, we designed rectangular distances to represent the specific direction(see Figure 3 ). the above processing method ensures the clear transmission of direction and distance information and the convenience of subsequent processing,

The merging module is the core of the algorithm, which identifies high-frequency RNA pairs by statistically analyzing the RNA pair counters generated during the preprocessing stage. For RNA pairs that meet the merging criteria, the system calculates the mean of their spatial position coordinates as the new position coordinates and updates relevant attributes. The merging process follows the strategy of "keep one line, delete one line" to ensure that data is not added or lost without reason.

### 4.1.3 Multiple iterations

The iteration control module manages the execution logic of the entire merge process, ensuring the stability and repeatability of the algorithm by dynamically creating intermediate output directories, maintaining merge history records, and controlling the number of iterations. The output data of each iteration will be used as input for the next iteration, forming a closed-loop optimization process.

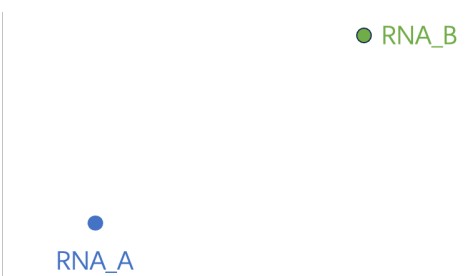

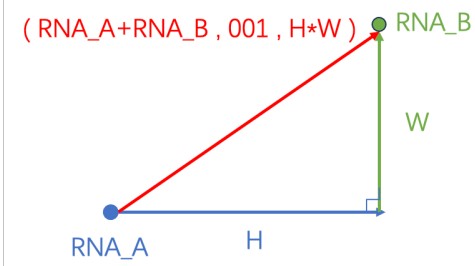

Figure 2: Double RNA Token Demonstration (Original)

Figure 3: Double RNA Token Demonstration (Tokenization)

### 4.2 TRAINING EMBEDDING

In order to better achieve RNA embedding vector training, we have designed a comprehensive implementation scheme aimed at integrating gene expression data with spatial location information to construct high-quality gene embedding representations. This scheme attempted the CBOW algorithm and Skip gram algorithm, considering their effectiveness and training time. The CBOW algorithm was chosen for training, and a comparison was made between randomly setting initial embeddings and loading one hot encoding as initial embeddings. One hot convergence was faster. The nearest neighbor data was calculated in memory, and the nearest neighbor gene list was extracted from the processed data to construct a training corpus. During the training process, the loss was dynamically monitored and the best embedding vector was saved. Additionally, a combination property regularization mechanism was applied to enhance the combination properties of embedded RNA. We also attempted to generate positional codes by combining gene spatial location information, but the effectiveness was not as good as adding positional information in subsequent training processes.

### 4.3 TRAINING MODELS AND PREDICTIONS

We trained a deep learning based cell type classification prediction model that integrates multiple strategies, including a BPE based gene sequence processing method that optimizes gene feature representation through two merging mechanisms; Innovatively introducing dual spatial position encoding to effectively integrate the spatial coordinate information of genes into the model; Implement multi-level data augmentation to improve model generalization ability; Using residual network architecture to construct a classifier, and combining techniques such as noise addition, regularization, Dropout, and gradient clipping to prevent overfitting. During the training process, adaptive learning rate adjustment and early stopping mechanism were applied, and the model was able to map gene expression data to multiple cell subtypes.

$$IPE = \sum_{i \in \text{x,y}} \text{positional encoding}(i) + \text{Initial gene embedding} \tag{1}$$

## 5 EXPERIMENTS

### 5.1 COMPARISON WITH STATE-OF-THE-ART METHODS AND ABLATION STUDIES ON STBPE

Moffitt et al. (2018) constructed a publicly available benchmark dataset using MERFISH, an imaging-based method for in situ cell type identification and mapping. This dataset comprises spatially resolved single cells from the mouse hypothalamic preoptic region, providing transcriptomic profiles and spatial coordinates for approximately one million cells across seven samples, along with high-confidence cell type annotations. To evaluate the cell annotation performance of STBPE, we conducted a comprehensive comparison against several state-of-the-art methods on this benchmark dataset. Specifically, we compared STBPE with STAGC and five recently released spatial clustering algorithms: DSCT, scDOT, TripletCell, TopACT, and Tosica.

Table 2: Results on Hypothalamic Preoptic Region Mouse 1 and Hypothalamic Preoptic Region Mouse 2. We report average ACC, Macro F1 Score (F1), as well as the Adjusted Rand Index (ARI). We highlight the best and second best mean.

| Model | Hypothalamic Preoptic Region Mouse 1 | | | Hypothalamic Preoptic Region Mouse 2 | | |
|---|---|---|---|---|---|---|
| | ACC | F1 | ARI | ACC | F1 | ARI |
| **STBPE** | **0.88 ± 0.01** | **0.86 ± 0.01** | **0.55 ± 0.08** | **0.86 ± 0.02** | **0.83 ± 0.03** | **0.53 ± 0.06** |
| Tosica | 0.86 ± 0.02 | 0.83 ± 0.03 | 0.32 ± 0.13 | 0.83 ± 0.03 | 0.81 ± 0.02 | 0.40 ± 0.09 |
| scDOT | 0.84 ± 0.02 | 0.81 ± 0.01 | 0.37 ± 0.10 | 0.83 ± 0.02 | 0.80 ± 0.03 | 0.41 ± 0.12 |
| DSCT | 0.83 ± 0.01 | 0.81 ± 0.03 | 0.29 ± 0.10 | 0.79 ± 0.04 | 0.76 ± 0.04 | 0.32 ± 0.11 |
| TripletCell | 0.81 ± 0.02 | 0.79 ± 0.03 | 0.17 ± 0.12 | 0.77 ± 0.04 | 0.76 ± 0.03 | 0.19 ± 0.09 |
| TopACT | 0.76 ± 0.03 | 0.74 ± 0.03 | 0.22 ± 0.13 | 0.73 ± 0.05 | 0.71 ± 0.06 | 0.26 ± 0.10 |

On both Hypothalamic Preoptic Region Mouse-1 and Mouse-2 datasets, STBPE consistently achieves the best performance across all metrics. It outperforms Tosica, scDOT, DSCT, TripletCell, and TopACT, with the highest ACC (0.88/0.86), F1 (0.86/0.83), and ARI (0.55/0.53), demonstrating superior accuracy and robustness in cell type annotation.

Table 3: Ablation study on embedding dimension and positional encoding.
(128/256/512 denote the size of token embeddings, and "Position" indicates the use of positional encoding.)

| Model | Hypothalamic Preoptic Region Mouse 1 | | | Hypothalamic Preoptic Region Mouse 2 | | |
|---|---|---|---|---|---|---|
| | ACC | F1 | ARI | ACC | F1 | ARI |
| STBPE-128 | 0.81 ± 0.02 | 0.80 ± 0.01 | 0.49 ± 0.05 | 0.80 ± 0.02 | 0.76 ± 0.02 | 0.49 ± 0.06 |
| STBPE-256 | 0.85 ± 0.01 | 0.84 ± 0.03 | 0.51 ± 0.06 | 0.84 ± 0.01 | 0.81 ± 0.02 | 0.51 ± 0.08 |
| STBPE-512 | 0.83 ± 0.02 | 0.82 ± 0.01 | 0.46 ± 0.10 | 0.81 ± 0.02 | 0.78 ± 0.04 | 0.47 ± 0.10 |
| STBPE-256-Position | **0.88 ± 0.01** | **0.87 ± 0.01** | **0.55 ± 0.08** | **0.86 ± 0.02** | **0.83 ± 0.03** | **0.53 ± 0.06** |

The ablation study shows that embedding size and positional encoding are critical to STBPE's performance. Among different embedding dimensions, 256 achieves the best overall results (ACC 0.88/0.86, F1 0.87/0.83, ARI 0.55/0.53). Larger (512) or smaller (128) embeddings perform slightly worse, while removing positional encoding leads to a clear drop across all metrics, confirming the importance of spatial information.

We further conducted an ablation study to examine the effect of varying the number of nearest neighbors during token embedding training. This analysis allows us to assess how local contextual information influences model performance. The detailed results are provided in Appendix E.

## 5.2 STBPE UNCOVERS CELL-TYPE-SPECIFIC SPATIAL TOKENS

In Figure 4, we visualize how STBPE transforms sparse RNA distributions into structured spatial tokens that correspond to distinct cell types and functional modules. For instance, astrocytes are captured by the token Aqp4+Cxcl14, highlighting the coupling between glial identity and immune chemokine signals; pericytes are represented by tokens such as Rgs5+Vtn and Flt1+Rgs5, reflecting their role in vascular stability and angiogenesis; excitatory cortical neurons are marked by Satb2+Tle4, while layer-specific projection neurons are characterized by Ptprk+Rorb; and microglia are revealed through Clq1+Slc30a3, indicating spatial interplay between immune regulation and synaptic activity. Moreover, novel tokens like Ndnf+Wipf3 and Parm1+Slc22a3 suggest previously underappreciated co-localization patterns, demonstrating that STBPE not only recovers known cell-type markers but also uncovers new spatial gene associations with potential biological relevance.

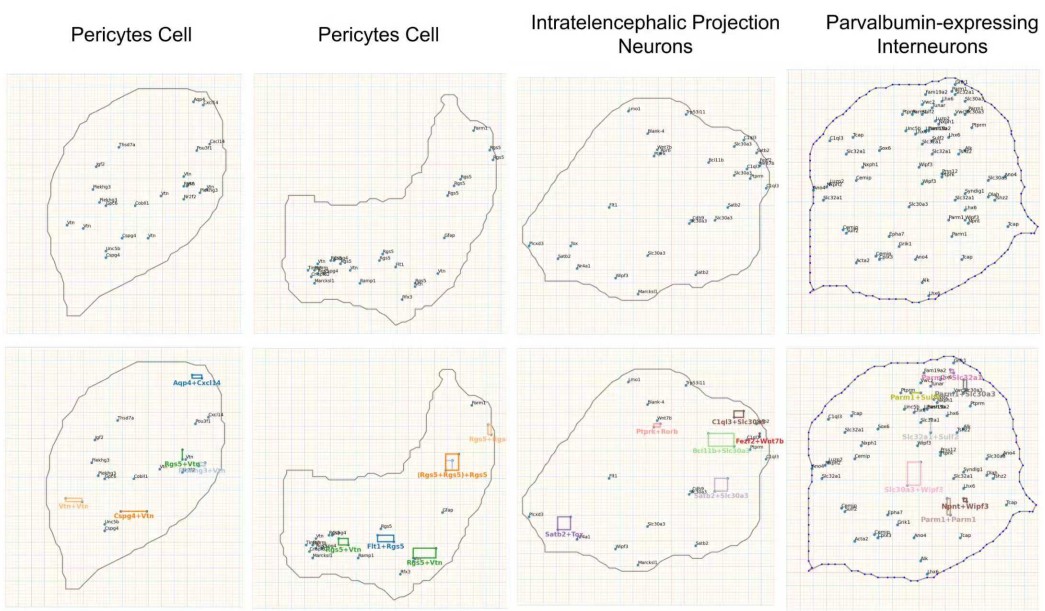

Figure 4: STBPE transforms sparse RNA distributions into structured spatial tokens linked to distinct cell types and functions.

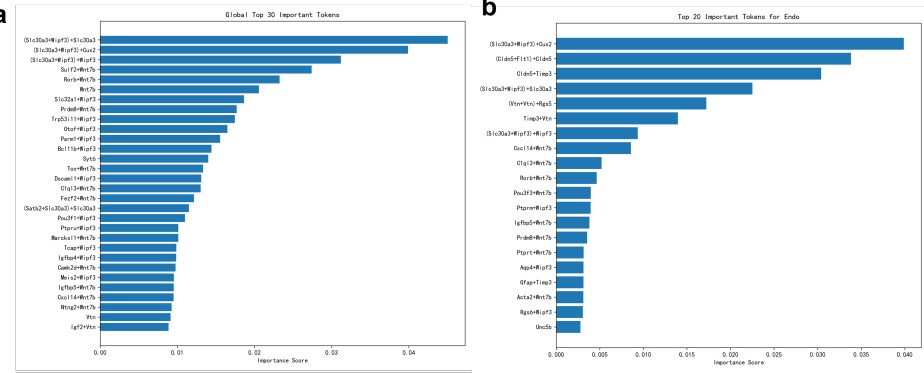

Figure 5: important tokens identified by IG analysis for four representative cell types (all cell and Endo cell)

### 5.3 INTERPRETABILITY ANALYSIS OF TOKEN IMPORTANCE USING THE INTEGRATED GRADIENTS (IG) METHOD

To better understand how STBPE contributes to cell classification, we perform an interpretability analysis using the Integrated Gradients (IG) method. This approach quantifies the contribution of each token to the model's predictions, allowing us to identify the most informative tokens and uncover biologically meaningful gene co-localization patterns.

Figure 5 presents the interpretability analysis of token importance using Integrated Gradients (IG). The top 30 most important tokens in the overall cell classification task (Figure 5.a) reveal that several higher-order tokens—such as (Slc30a3+Wipf3)+Slc30a3 and (Slc30a3+Wipf3)+Cux2—achieve the highest importance scores, surpassing most pairwise or single-gene tokens. This indicates that compositional tokens integrating multiple RNA signals capture richer spatial and functional contexts, which are critical for distinguishing fine-grained cell identities. For endothelial cells (Endo) (Figure 5.b), tokens such as (Cldn5+Flt1)+Cldn5 and Cldn5+Timp3 dominate, reflecting the role of Cldn5 and Flt1 as canonical endothelial markers of vascular integrity and angiogenic signaling. These

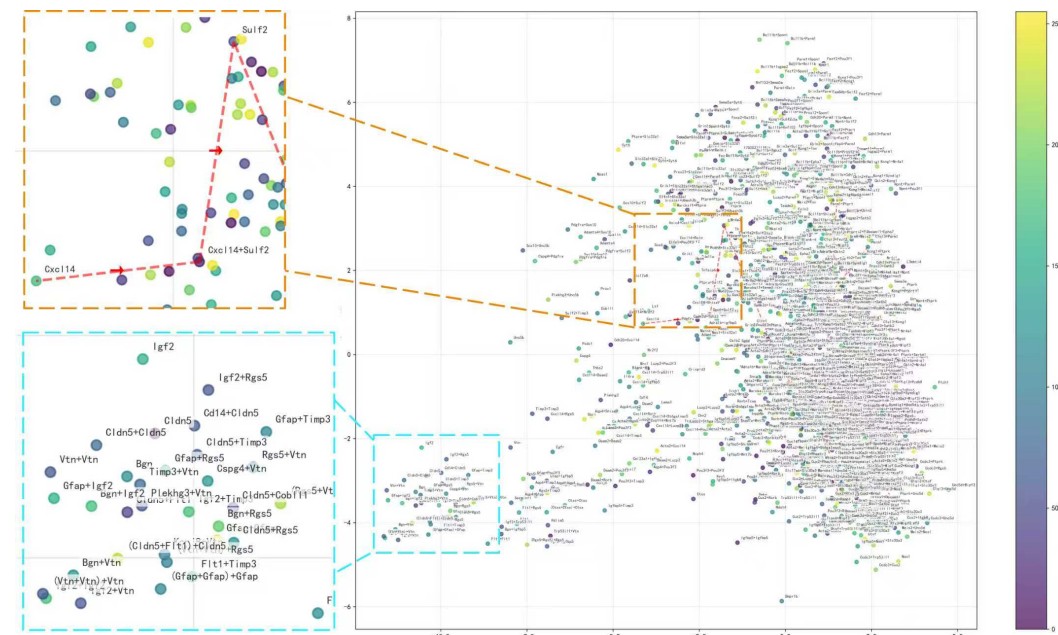

Figure 6: 2D visualization reveals the spatial proximity of tokens with high biological relevance.

analyses demonstrate that higher-order tokens capture biologically coherent co-localization patterns, integrating cell-type markers with regulatory and signaling genes, thereby offering mechanistic insights into cellular identity. Additional examples of cell-type-specific key tokens are provided in the Appendix G.

## 5.4 VISUALIZATION OF SPATIAL TOKEN EMBEDDINGS

To further examine whether STBPE embeddings capture biologically meaningful structures(Figure 6), we projected the 256-dimensional RNA token embeddings into a 2D space for visualization. Figure 6 presents the resulting distribution, where tokens with functional or spatial relevance cluster together. The orange zoom-in panel highlights a trajectory from Cxcl14 to Cxcl14+Sulf2 to Sulf2, illustrating how merged tokens naturally bridge their component genes and reflect the coupling between immune chemokine signaling and developmental regulation. The blue zoom-in panel shows that tokens with related biological semantics tend to cluster, such as (Cldn5+Flt1)+Cldn5 and Clnd5+Timp3 (endothelial barrier and remodeling), or Rgs5+Vtn and Gfap+Timp3 (vascular and pericyte–astrocyte interactions). These results demonstrate that the embedding space organizes tokens into biologically coherent modules, with higher-order tokens serving as bridges that encode multi-gene co-localization patterns.

# 6 CONCLUSION

In this work, we introduced STBPE, a generative pre-training framework that transforms subcellular-resolution RNA spatial distributions into spatial tokens while explicitly preserving relative positional relationships such as distances and angles. Through comprehensive experiments, we demonstrated that STBPE consistently improves cell type annotation accuracy, generalizes robustly across datasets, and uncovers biologically meaningful co-localization patterns that extend beyond conventional markers. Furthermore, interpretability analyses reveal that higher-order tokens capture coherent regulatory modules, offering new insights into the molecular basis of cellular identity and disease mechanisms. Looking ahead, STBPE provides a foundation for constructing multimodal pre-trained models that integrate spatial transcriptomics with other omics and imaging modalities, thereby opening new avenues for biological discovery and translational medicine.

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

## A   ETHICS STATEMENT

This work adheres to the ICLR Code of Ethics. We are committed to conducting research with integrity, transparency, and responsibility. Specifically, we ensure that our study contributes to scientific excellence and the broader well-being of society, avoids harm, and respects privacy and confidentiality. We acknowledge all contributions fairly, uphold fairness and inclusivity, and take action to prevent discrimination. Furthermore, we strive to present our methods and results with honesty and reproducibility, and we carefully consider the potential societal and environmental impacts of our findings.

## B   THE USE OF LARGE LANGUAGE MODELS (LLMS)

In this work, we made use of large language models (LLMs) to assist with writing and refinement, as well as for information retrieval and discovery (e.g., identifying relevant prior work). The use of LLMs was limited to supporting tasks that enhance clarity and efficiency in manuscript preparation and literature exploration; all core scientific contributions, including conceptualization, experimental design, model development, data analysis, and interpretation of results, were conducted independently by the authors. We acknowledge the assistance of LLMs explicitly to ensure transparency and uphold research integrity in accordance with the ICLR Code of Ethics.

## C   REPRODUCIBILITY STATEMENT

Reproducibility Statement. We have made every effort to ensure that our results are reproducible. The complete code and configurations are open-sourced at https://github.com/Guolujiale/STBPE (including training/validation/test scripts, environment files, data download and preprocessing instructions, logging and visualization tools). Details of the model and algorithms are provided in Sections 3 and 4 of the main text.

# D  ALGORITHM 1 STBPE (SPATIAL TRANSCRIPTOMICS BYTE-PAIR-ENCODING)

---

**Algorithm 1** STBPE (Spatial Transcriptomics Byte Pair Encoding)

---

**Require:** Cell database (RNA groups by cell), Number of merges $K$
**Ensure:** Vocabulary $V$ (initial + merged RNAs), Merge rules $R$ (with spatial info)

```
 1: for each cell in database do
 2:     Sort RNAs                                          ▷ Prevent merge ambiguity
 3: end for
 4: V ← {unique RNAs in database}
 5: freq ← {RNA frequency counts}
 6: R ← {id, RNA, position, H*W}
 7: for i ← 1 to K do
 8:     pair_freq ← {}
 9:     for each cell in database do
10:         for each RNA r in cell do
11:             n ← FindNearestRNA(r, cell)              ▷ Euclidean distance
12:             pair ← (r, n)
13:             pair_freq[pair] ← pair_freq.get(pair, 0) + freq[r]
14:         end for
15:     end for
16:     if pair_freq is empty then break
17:     end if
18:     best ← arg max_{p∈pair_freq} pair_freq[p]
19:     v ← Concatenate(best[0], best[1])
20:     V.add(v)
21:     R[best] ← {order : i, names : best,
22:         orientation : CalculateOrientation(best),
23:         size : CalculateSize(best)}
24:     freq ← {rna : f | (rna, f) ∈ freq, rna ∉ best}
25:     freq[v] ← pair_freq[best]
26: end for
27: return V, R
```

---

# E   ABLATION ON NEAREST NEIGHBOR SIZE IN TOKEN EMBEDDING

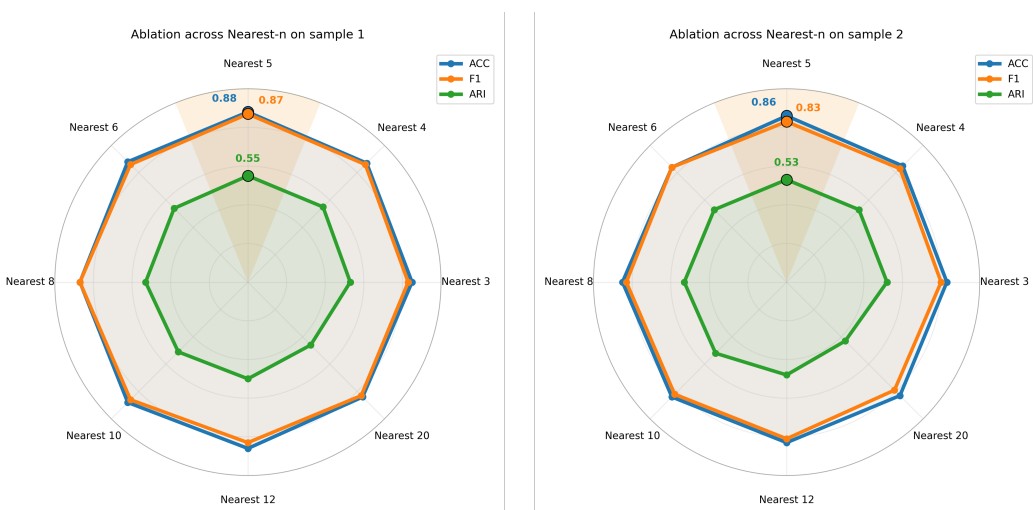

Figure 7: Enter Caption

Impact of varying the number of nearest neighbors during token embedding training. We evaluated neighbor sizes of 3, 4, 5, 6, 8, 10, 12, and 20 on two benchmark samples. The results show that increasing the number of neighbors generally enriches contextual information and improves model performance, with Nearest-5 achieving the best overall scores (Sample 1: ACC = 0.88, F1 = 0.87, ARI = 0.55; Sample 2: ACC = 0.86, F1 = 0.83, ARI = 0.53). However, when the neighbor size becomes too large, the benefit diminishes and performance declines, likely due to informative signals being diluted across excessive neighbors.

# F  OVERVIEW OF THE MOUSE HYPOTHALAMIC PREOPTIC REGION DATA

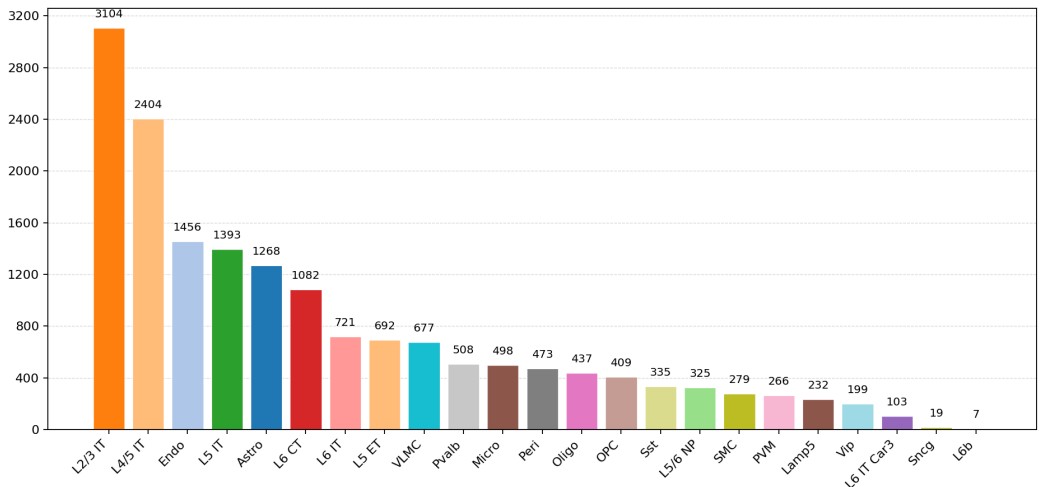

Figure 8: Cell type composition in Sample 1 of the mouse hypothalamic preoptic region dataset.

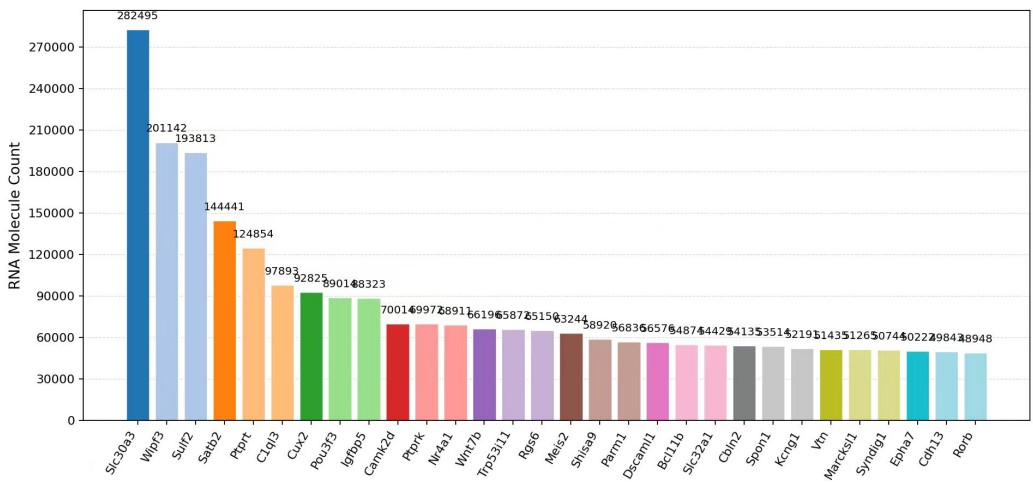

Figure 9: Top 30 most abundant RNA species in Sample 1.

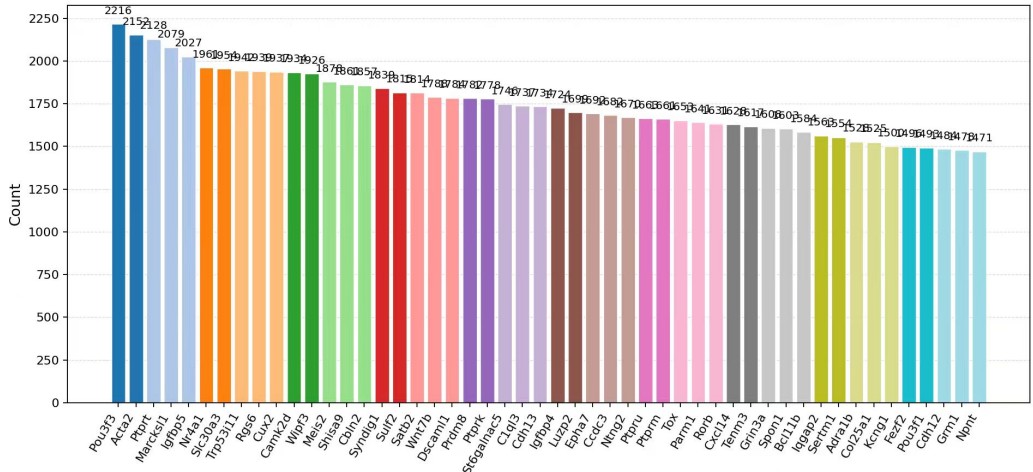

Figure 10: Top 50 most frequent tokens after STBPE tokenization in Sample 1.

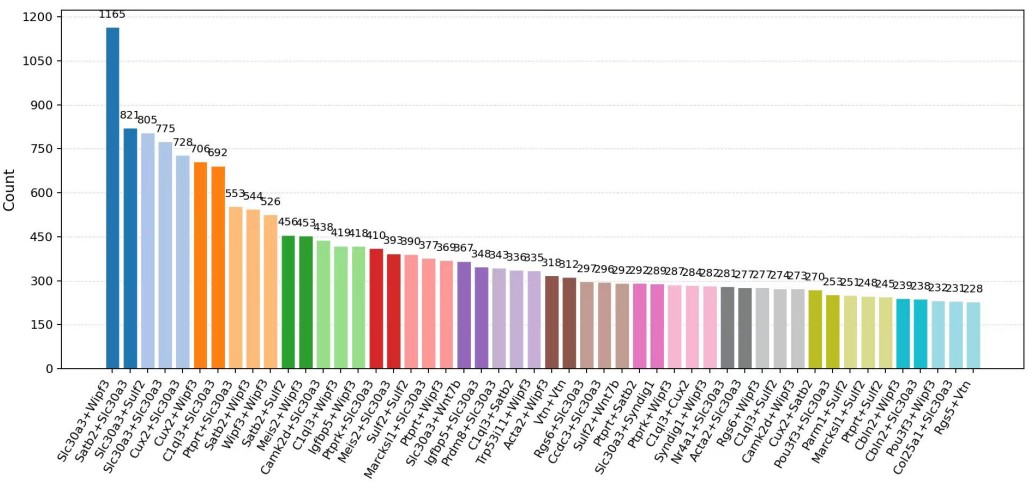

Figure 11: Top 30 most frequent binary tokens after STBPE tokenization in Sample 1.

# G APPENDIX D CELL-TYPE-SPECIFIC STRONGLY ASSOCIATED TOKENS IDENTIFIED BY IG ANALYSIS

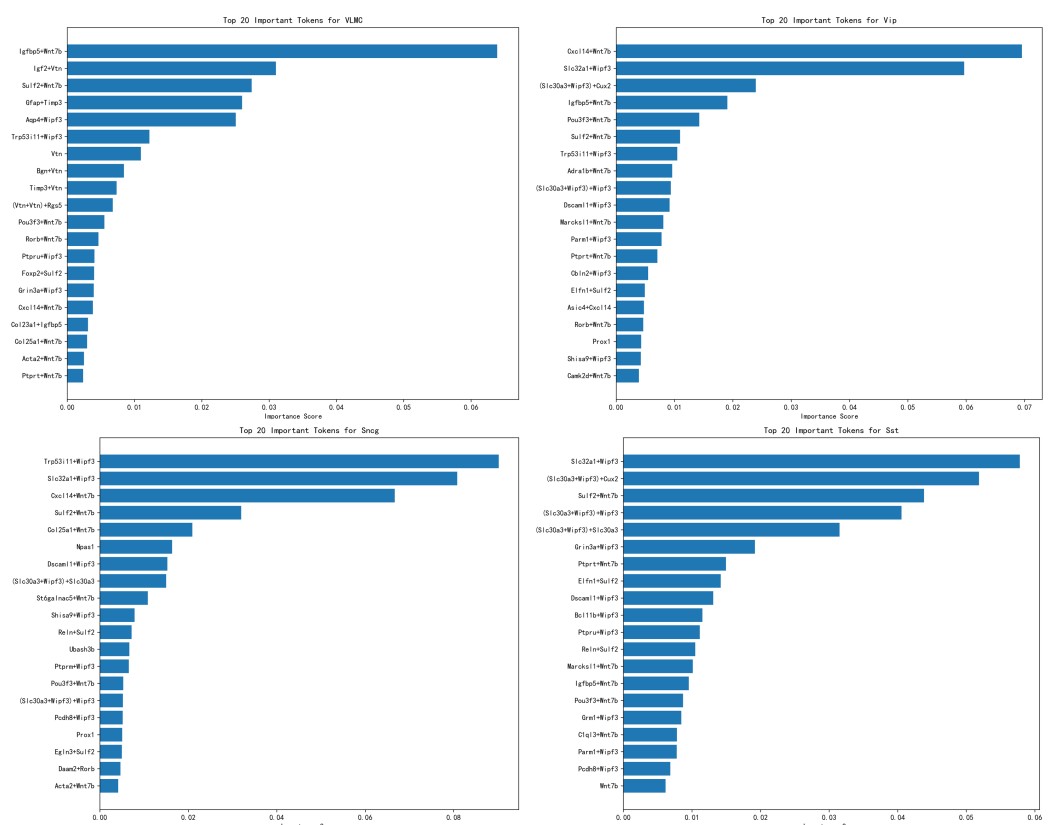

Figure 12: Key tokens of neuron-associated cells reveal cell-type-specific molecular co-localization patterns

Top 20 important tokens identified by IG analysis for four representative cell types (VLMC, Vip, Sncg, and Sst). The IG analysis highlights cell-type-specific token importance patterns. For VLMCs (vascular and leptomeningeal cells), tokens such as Igfbp6+Wnt7b and Igf2+Vtn dominate, reflecting their roles in extracellular matrix organization and vascular support. For Vip interneurons, tokens including Cxcl14+Wnt7b and (Slc30a3+Wipf3)+Cux2 are enriched, suggesting the interplay between chemokine signaling and neuronal differentiation programs. For Sncg neurons, tokens such as Trp53i11+Wipf3 and Slc32a1+Wipf3 appear as key drivers, pointing to regulatory and synaptic scaffolding factors crucial for neuronal identity. For Sst interneurons, tokens like Slc32a1+Wipf3 and (Slc30a3+Wipf3)+Cux2 are most important, indicating a convergence of synaptic regulation and transcriptional control in defining inhibitory interneuron subtypes. Collectively, these results demonstrate that IG analysis reveals biologically coherent and cell-type-specific multi-gene token patterns, offering interpretable insights into the molecular basis of cell identity.

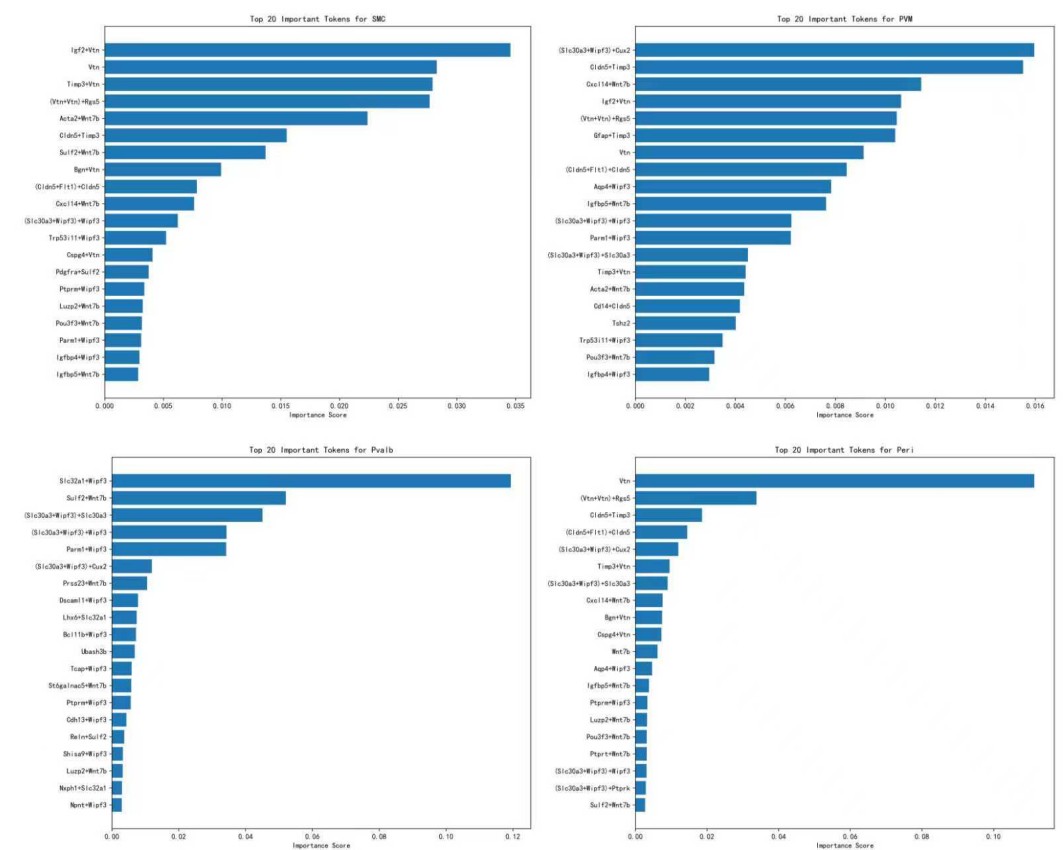

Figure 13: Top 20 important tokens identified by IG analysis for four representative cell types (SMC, PWM, Pvalb, and Peri).

Description. The IG analysis highlights key token patterns that are highly specific to vascular- and neuron-associated cells. For smooth muscle cells (SMC), tokens such as Igf2+Vtn and Vtn+Vtn dominate, reflecting their central role in extracellular matrix organization, vascular con- tractility, and structural support. For perivascular macrophages (PWM), important tokens include (Slc30a3+Wipf3)+Cux2 and Cldn5+Timp3, which indicate interactions between immune regula- tion, vascular barrier integrity, and transcriptional control. In parvalbumin interneurons (Pvalb), tokens such as Slc32a1+Wipf3 and Sulf2+Wnt7b are most important, emphasizing the role of in- hibitory synaptic transmission and Wnt-mediated developmental pathways. For pericytes (Peri), tokens like Vtn and (Vtn+Vtn)+Rgs5 stand out, underscoring their function in vascular stability and vessel–matrix interactions. Collectively, these results suggest that STBPE captures biologically co- herent, cell-type-specific co-localization patterns, integrating canonical markers with regulatory and signaling genes to reveal deeper insights into vascular and neuronal cell identities.

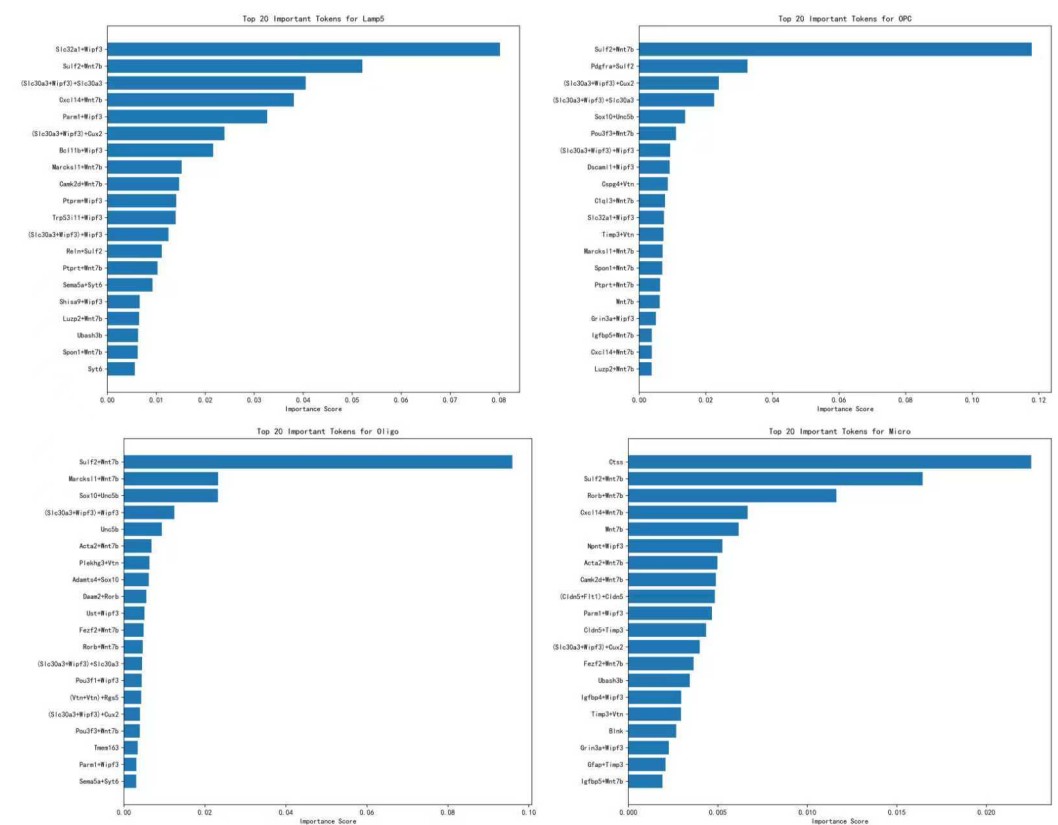

Figure 14: Top 20 important tokens identified by IG analysis for four representative cell types (Lamp5, OPC, Oligo, and Micro).

The IG analysis reveals token patterns reflecting the molecular basis of glial and immune-related cell identities. For Lamp5 interneurons, tokens such as Slc32a1+Wipf3 and Sulf2+Wnt7b dominate, highlighting the importance of inhibitory synaptic regulation and developmental signaling in these specialized interneurons. For oligodendrocyte precursor cells (OPC), key tokens include Sulf2+Wnt7b and Pdgfra+Sulf2, consistent with their roles in differentiation and developmental signaling pathways. For oligodendrocytes (Oligo), the most important tokens are Sulf2+Wnt7b and Marcks1+Wnt7b, reflecting the integration of Wnt-mediated differentiation signals with membrane-associated scaffolding crucial for myelination. For microglia (Micro), tokens such as Ctss, Sulf2+Wnt7b, and Rorb+Wnt7b dominate, pointing to immune and proteolytic functions (Ctss), developmental signaling (Wnt7b), and transcriptional regulation (Rorb) as key drivers of microglial identity. Collectively, these results demonstrate that STBPE captures biologically coherent and cell-type-specific co-localization modules across inhibitory neurons, glial cells, and immune cells, thereby enhancing interpretability of spatial transcriptomics data.

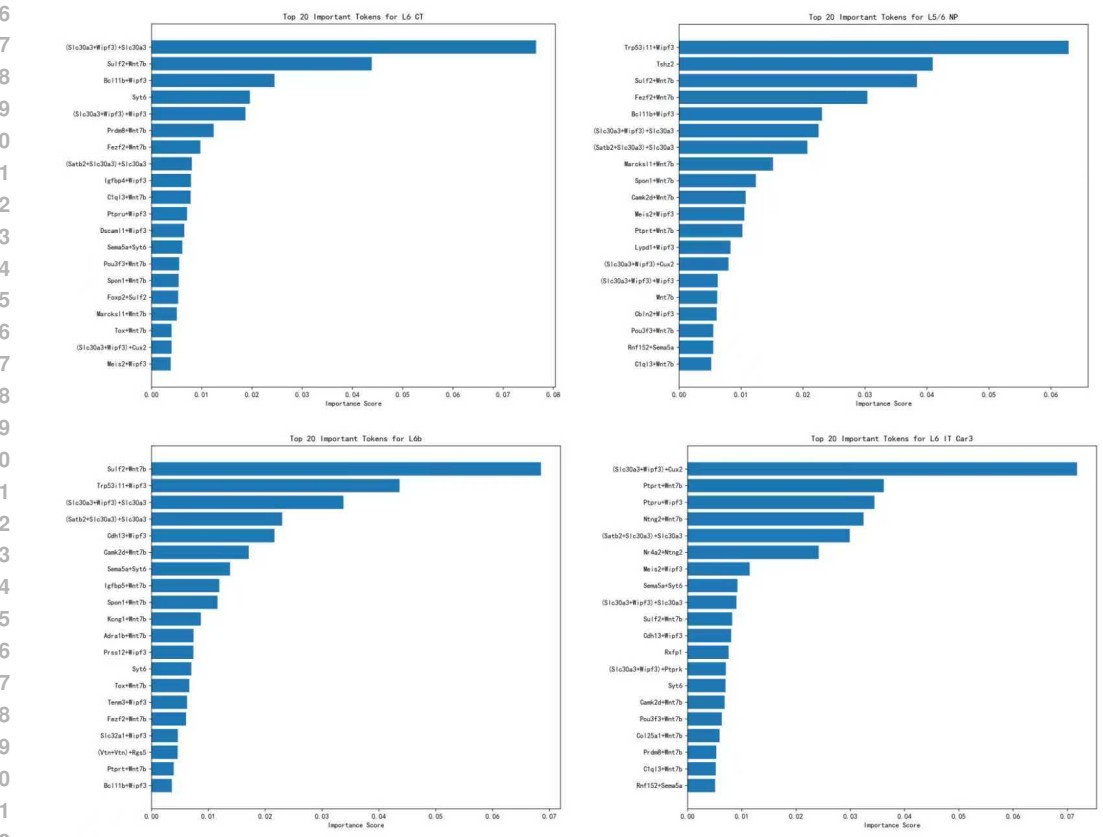

Figure 15: Top 20 important tokens identified by IG analysis for four excitatory neuron subtypes (L6 CT, L5/6 NP, L6b, and L6 IT Car3).

The IG analysis highlights subtype-specific molecular signatures within deep-layer excitatory neurons. For L6 corticothalamic neurons (L6 CT), tokens such as (Slc30a3+Wipf3)+Slc30a3 and Sulf2+Wnt7b dominate, reflecting the importance of synaptic scaffolding and Wnt signaling in corticothalamic specification. For L5/6 near-projecting neurons (L5/6 NP), tokens like Trp53i11+Wipf3 and Tnc+Z (with Sulf2+Wnt7b also enriched) indicate regulatory and extracellular matrix–associated signals essential for projection neuron identity. For L6b neurons, key tokens include Sulf2+Wnt7b and Trp53i11+Wipf3, pointing to the convergence of developmental cues and transcriptional regulation in this distinct late-born cortical population. For L6 IT Car3 neurons, tokens such as (Slc30a3+Wipf3)+Cux2 and Ptpn4+Wnt7b are most important, highlighting transcriptional regulators and Wnt signaling modules that underlie intratelencephalic projection. Collectively, these results suggest that STBPE uncovers biologically coherent, subtype-specific co-localization patterns, providing mechanistic insights into the molecular programs that diversify cortical excitatory neurons in deep layers.

