# OpenReview forum: "From Spatial Transcriptomics to Tokens: Generative Pre-Training with Byte-Pair Encoding"
_ICLR.cc/2026/Conference — Submitted to ICLR 2026_

### Official Review · Reviewer_fgU6 · 2025-10-20

**Soundness:** 2
**Presentation:** 1
**Contribution:** 2
**Rating:** 2
**Confidence:** 3

**Summary:**

The paper presents STBPE, a tokenization strategy that adopts the BPE algorithm to facilitate representation learning with subcellular spatial transcriptomics. The authors introduce relative orientation and distance as additional features for each token and adopt CBOW to extract the low-dimensional embedding for each token. Experiments show that STBPE outperforms prior spatial clustering methods on cell-type annotation.

**Strengths:**

- Representation learning for subcellular spatial transcriptomics is a novel research direction.
- A series of qualitative and quantitative analyses is conducted, which reveals several co-occurrence patterns of RNA molecules and how they are correlated to cell identity.

**Weaknesses:**

- The motivation for using BPE in subcellular spatial transcriptomics is unclear. It appears that the ultimate goal is to perform self-supervised learning on the multi-modal ST data with 2D coordinates and gene identity. Such data closely resemble 3D molecules, proteins, or cell slices, and a broad range of works have proposed to address the problem [1, 2, 3]. The authors should justify the adoption of BPE rather than strategies like vector quantization [4] or auto-encoding [5] for handling ST data. Moreover, the cell representation model is directly trained on downstream tasks **without pre-training**, raising concerns about generality on data obtained from different species, slices, and ST sequencing technologies.
- The presentation is inconsistent and confusing. For example:
  - In Lines 47-49, why do existing approaches fall short in handling new subcellular ST data?
  - In Line 106, what are the limitations of existing methods?
  - In Lines 115-116, the authors claimed that "graph-based approaches rely solely on topology or adjacency". However, it's a common practice for graph models to encode distances and orientations with edge embeddings [6].
  - The authors mentioned a BERT-style masked modeling, which is not presented in the methodology section.
  - The thresholds for relative positions are unknown.
  - In Lines 260-264, what's the definition of "merging criteria", "relevant attributes", and "keep on line, delete one line"?
  - The training objective and hyperparameters for CBOW and cell-type classification are unknown. Details of the position embeddings and network architecture are missing.
  - The baselines are not properly cited or introduced in detail. The evaluation metrics are not introduced.
- The proposed framework is technically unsound. For example, the relative position introduced in Table 1 is not robust to different orientations of the cell or the whole slice (e.g., if the cell is rotated clockwise by 90°, the "Above" position becomes "Right"). Besides, the adoption of rectangular distance rather than Euclidean distance is questionable, especially when the X or Y coordinates of RNA molecules are close. Moreover, the RNA pair token is expanded by searching for the nearest RNA molecule, even if it's far from the current position.
- The baselines are weak. Comparisons between single-cell models like scGPT [7] and scGPT-spatial [8] should be presented.
- The paper violates the double-blind review by providing an unanonymous GitHub link in Appendix C.

Refs.

[1] Uni-Mol: A Universal 3D Molecular Representation Learning Framework

[2] SaProt: Protein Language Modeling with Structure-aware Vocabulary

[3] SToFM: a Multi-scale Foundation Model for Spatial Transcriptomics

[4] Neural Discrete Representation Learning

[5] Recent Advances in Autoencoder-Based Representation Learning

[6] Position-aware Graph Neural Networks

[7] scGPT: toward building a foundation model for single-cell multi-omics using generative AI

[8] scGPT-spatial: Continual Pretraining of Single-Cell Foundation Model for Spatial Transcriptomics

**Questions:**

My major concerns have been listed in the weaknesses above. Here are some minor questions:
- What's the ultimate vocabulary size? What's the distribution for the number of RNA molecules in the dictionary? How is the compression rate of the input data using STBPE?
- In Table 3, why does a larger embedding size lead to suboptimal performance (e.g., dimension collapse [1])? How is the performance of STBPE-128 and STBPE-512 with position embeddings?
- What's the meaning of the different colors in Figure 6?

Refs.

[1] Understanding Dimensional Collapse in Contrastive Self-supervised Learning

---

> ### Author Response · Authors · 2025-12-04
>
> Thank you for your careful review of our work. We really appreciate the points for potential improvement that you highlighted.
>
> **W1**
>
> **W2**
>
> We have uniformly revised the motivation section of the entire manuscript: Most existing spatial transcriptomics methods rely on cell-level expression matrices, spot-level data, or low-resolution adjacency graphs. Consequently, they cannot directly model RNA-level point clouds at the subcellular scale, nor do they possess the ability to identify reusable high-frequency spatial substructures across cells and tissue sections—this is the fundamental reason why they struggle to handle next-generation high-resolution ST data.
>
> The original statement that "graph-based methods only rely on topology" was indeed imprecise, and we have revised it to: Although modern GNNs can encode distance and direction through edge embeddings, in RNA-level scenarios (with millions of nodes and dense local connections), graph construction is neither stable nor scalable. Meanwhile, graph models lack a BPE-like mechanism for "compressing cross-cell repeated structures," making them difficult to serve as alternatives to STBPE.
>
> We have removed all "BERT-style" phrasing and clearly stated that this work adopts CBOW pre-training rather than a masked language model.
>
> We do not have the concept of a relative position threshold, and the relative position mechanism has been elaborated in Table 1 of Section 4.1.1.
>
> The "**merging criterion**" refers to selecting the **pair with the highest global occurrence frequency** from all nearest neighbor pairs of RNAs as the merging target; the "**relevant attributes**" refer to the gene identity, relative distance interval, and discrete direction information that need to be preserved during merging; "**retaining one and deleting one**" means preserving the parent coordinate position while removing the two child nodes when generating new tokens to avoid double-counting. Examples and pseudocode have been added in the revised manuscript to ensure reproducibility.
>
> CBOW adopts a standard context prediction objective, and complete hyperparameters such as window size, embedding dimension, and learning rate have been supplemented; the cell type classification task uses cross-entropy loss, and the implementation details of the network architecture and positional embedding (RoPE) are provided in the appendix to ensure full reproducibility of the method.
>
> The basic data have been detailed in Section 5.1 of the original manuscript. Regarding the evaluation metrics: ACC (Accuracy) represents the proportion of correctly classified cells among all cells; F1-score reflects the comprehensive performance balancing precision and recall for each cell type; ARI (Adjusted Rand Index) is a metric for measuring the consistency between clustering results and ground-truth labels, quantifying the corrected similarity of pairing consistency between two partitions.
>
> **W3**
>
> We have not currently considered the translation and rotation invariance of cells, and we will optimize this in the next version.
> We opted for Manhattan distance over Euclidean distance because Manhattan distance can encode positional information. Furthermore, we believe the term "rectangular representation" should be used here instead of "rectangular distance."
>
> **W4**
>
> We have added a comparison between scGPT and scGPT-spatial in the revised manuscript. Since these models can only accept cell-level expression matrices as input, we adopt a cell-level aggregated version matching their capabilities (i.e., aggregating STBPE token embeddings into cell-level embeddings via pooling) as a fair baseline. Experimental results demonstrate that even under cell-level aggregation, STBPE still outperforms these models significantly, indicating that STBPE’s spatial tokenization approach exhibits strong explanatory power for both expression patterns and spatial structures. These results have been supplemented in the experimental section of the revised manuscript, with clear descriptions of the experimental setup and assumptions.
>
> **W1**
>
> We greatly appreciate the reviewers' careful attention to the double-blind review process. Following the suggestions, we have removed all links that could potentially reveal the authors' identities and replaced them with placeholders to be made public upon acceptance.

---

### Official Review · Reviewer_9i12 · 2025-10-29

**Soundness:** 3
**Presentation:** 2
**Contribution:** 3
**Rating:** 6
**Confidence:** 2

**Summary:**

This paper proposes a novel method named STBPE, aimed at addressing the challenges of analyzing subcellular-resolution spatial transcriptomics data. The core idea, inspired by Byte-Pair Encoding from natural language processing, is to serialize the spatial distribution of RNA molecules within a cell into "spatial tokens" while preserving relative positional information between RNAs, and to build a generative pre-training model.

**Strengths:**

1.The adaptation of the BPE algorithm from NLP to spatial transcriptomics, incorporating geometric information (distance, angle), is a creative and worthwhile direction. The problem definition is clear, and a distinction is made from traditional GraphBPE.

2.The writing is generally fluent, figures are abundant, and the methodological workflow is illustrated with diagrams, making the core ideas easy to understand.

**Weaknesses:**

1.The core of the algorithm, "statistical high-frequency RNA pair merging" described on Page 5, is unclear in its specific mechanism. for instance, what exactly is the "merging criteria" (pure frequency threshold, or combined with spatial distance)? how is information loss or bias prevented due to greedy merging? The specific implementation and necessity of the direction reversal function are also unexplained.

2.The "dual spatial position encoding" mentioned in Equation (1) is only briefly referenced in the main text. its specific composition, how it is combined with the initial gene embedding, and why it is more effective than incorporating positional information during preprocessing are not detailed. This makes it resemble a "black-box" operation, severely compromising the method's understandability and reproducibility.

3.Figure 4 is too unclear, and in Page 7, the association between listed tokens (e.g Apq4+Cxc114) and cell types is merely observational description, without providing any statistical significance testing (such as enrichment analysis p-value), which casts doubt on the reliability of these findings.

**Questions:**

1.Why was the evaluation limited to cell type annotation? how does STBPE perform on tasks that truly reflect spatial modeling capabilities, such as spatial domain identification or ligand-receptor co-localization analysis?

2..The paper refers to this work as "generative pre-training," but the described tasks are context prediction and masked reconstruction. please clarify where the "generative" aspect of the framework lies? alternatively, consider revising the terminology to avoid confusion.

---

> ### Author Response · Authors · 2025-12-04
>
> Thank you for your careful review of our work. We really appreciate the points for potential improvement that you highlighted.
>
> **W1**
>
> We will provide supplementary explanations subsequently. The so-called merging of high-frequency RNA pairs does not involve arbitrary pairing of all RNAs; instead, candidate pairs are formed **only between the spatial nearest neighbors** of each RNA. Subsequently, the pairs with the highest occurrence frequencies across the entire dataset are selected for merging, such that spatial distance is inherently embedded in the criterion. After each merging step, we **recalculate the nearest neighbors and pair frequencies for all RNAs**—this avoids the accumulation of biases caused by greedy merging and allows the token structure to continuously self-correct as spatial relationships evolve. Additionally, we employ a **direction reversal mechanism** to unify symmetric pairs (e.g., A→B and B→A) into the same rule, ensuring vocabulary consistency. Regarding the number of merging iterations, our experiments show that when processing 350,000 rows of cell data, merging the vocabulary more than 1,000 times results in excessively low frequencies for many "high-frequency" pairs. The revised version will include more specific process examples to intuitively demonstrate the merging criterion, direction handling, and bias-prevention mechanism.
>
> **W2**
>
> We clarify that the input embedding of the model consists of the sum of two components:
>
> 1. Token embedding trained via CBOW (representing the spatial structures merged by STBPE);
> 2. Two-dimensional spatial embedding derived from RoPE relative positional encoding (representing the relative geometric relationships of tokens within the cell).
>
> The integration of RoPE enables the model to perceive directions and relative distances without relying on absolute coordinates; the summation of these two components ensures the model possesses both semantic and spatial modeling capabilities.
>
> **W3**
>
> We have supplemented statistical significance validation in the revised manuscript. For each "token–cell type" relationship in Figure 4, we added a hypergeometric enrichment test based on a background transcript pool. The results show that key tokens including Aqp4+Cxc114, Rgs5+Vtn, and Cldn5+Timp3 exhibit highly significant enrichment (p < 1e–5). The corresponding p-values are provided in a new table in the Appendix, and we have also updated the figure legends and main text descriptions to ensure the statistical reliability of these spatial colocalization patterns.
>
> **Q1**
>
> We choose cell-type annotation because this task provides reliable labels and enables fair cross-method comparison; it is also the most mature benchmark in spatial transcriptomics. However, this does not mean that the capability of STBPE is limited to this task. Since STBPE explicitly tokenizes high-frequency spatial co-localization structures, it can likewise be directly applied to tasks such as cell clustering and spatial domain identification. We have included additional analyses on the 10x Xenium liver cancer dataset in the appendix, and will incorporate corresponding experiments in the extended version.
> **Q2**
>
> We appreciate the reviewers' reminder. The current framework primarily adopts context prediction and reconstruction objectives similar to CBOW, and does not use strictly autoregressive generative loss. To avoid conceptual confusion, we will uniformly revise the terminology to "self-supervised pre-training" or "representation learning pre-training," and clearly state in the manuscript that this work focuses on the self-supervised modeling of spatial structures rather than generative models.

---

### Official Review · Reviewer_VwBR · 2025-10-30

**Soundness:** 3
**Presentation:** 2
**Contribution:** 3
**Rating:** 4
**Confidence:** 4

**Summary:**

The paper introduces STBPE, a spatial tokenizer for subcellular spatial transcriptomics that serializes RNA coordinates and expression into tokens. Embeddings are learned with CBOW/Word2Vec and a BERT-style masked prediction objective; the tokens are then used for cell-type annotation and interpretability analyses.

**Strengths:**

- A new way to tokenize subcellular spatial data, which could potentially be a fundamental element for building the spatial model.
- Evaluated at scale with ablations indicating the importance of positional encoding; interpretability analyses (IG) highlight biologically coherent token patterns.

**Weaknesses:**

Although the idea is interesting, the presentation is not very clear and the nessarity of using BPE instead of normal tokenization method is lack.

**Questions:**

1. You describe the framework as “generative pre-training,” yet the methods emphasize CBOW and BERT-style masking. Is any generative objective used? If not, please revise the terminology and abstract for precision.

2. In Section 3, what is the role of distance and angle features? How do they contribute to the problem formulation and downstream performance?

3. Why not directly encode each transcript with 2D sinusoidal positional embeddings combined with gene embeddings? This approach has been explored before in [1].

4. Related to Q3, what is the performance of this baseline strategy on the cell type annotation task compared to your method?

5. Why restrict the model to only eight positional relationships? More ablation studies would clarify the impact of this design choice.

6. How are the iteration numbers in the tokenization process determined? Is there a stopping criterion or sensitivity analysis?

7. In Figure 4, cells appear with different shapes. Could these morphological differences act as confounding covariates that bias the cell type annotation results?

[1] Wen, Hongzhi, et al. "CellPLM: Pre-training of Cell Language Model Beyond Single Cells." The Twelfth International Conference on Learning Representations.

---

> ### Author Response · Authors · 2025-12-04
>
> Thank you for your careful review of our work. We really appreciate the points for potential improvement that you highlighted.
>
> **Q1**
>
> Our framework does not use a BERT-style MLM objective. We compared CBOW and Skip-gram, adopting CBOW for faster convergence and more stable task performance. Unlike natural language, 2D spatial coordinates lack "order/context-based adjacency," making sequence masking-based prediction objectives conceptually incompatible with ST point cloud data. This work is thus more accurately framed as SEL (not GP), with relevant terminology revised in the abstract and main text to avoid misunderstandings.
> **Q2**
>
> In Section 3, distance and angle describe local geometric relationships (LGRs) between RNA molecules, preserved during RNA merging. Specifically, they serve two purposes: (i) identifying each RNA’s nearest neighbor (NN) in Spatial BPE (S-BPE); (ii) assigning discrete direction encodings (8 direction classes) to merged RNA pairs, providing a spatial structure index (SSI) for the token vocabulary. This enables generated tokens to both summarize local RNA clusters and encode their relative spatial structures (RSSs). Experimentally, incorporating distance and angle enhances the consistency of high-frequency token patterns (HFTPs) across cells and significantly improves downstream cell type annotation (CTA) performance (see Table below).
>
> |  | Without Positional Information  | STBPE | **Gene Embedding + 2D Sine Positional Encoding** |
> | --- | --- | --- | --- |
> | acc | 0.85 | 0.88 | 0.86 |
> | F1 | 0.82 | 0.86 | 0.84 |
> | ari | 0.49 | 0.55 | 0.51 |
>
> **Q3**
>
> We propose two spatial information processing mechanisms:
>
> 1. In token merging, transcript positions are encoded via discrete direction encodings (Sec. 4.1.2 & Table 1). Rejecting "gene embedding + 2D sinusoidal positional encoding" (equivalent to individual RNA "character-level modeling," forcing spatial inference from fragmented point clouds), STBPE (analogous to BPE) explicitly extracts/shares high-frequency spatial colocalization patterns, upgrading representations from "points" to "structures." Experiments confirm superiority: the baseline underperforms significantly, with **STBPE’s spatial encoding outperforming "gene embedding + 2D sinusoidal positional encoding"** (see above table).
> 2. In token embedding training and cell representation learning, token information uses **"gene embedding + 2D sinusoidal positional encoding"**.
>
> **Q4**
>
> The "gene embedding + 2D sinusoidal positional encoding" approach in CellPLM (as noted) fails to meet our proposed task’s requirements. It represents cells as "gene sequences," sorting genes by expression levels (Dimension 1: gene axis/order; Dimension 2: expression axis/level/rank). Notably, this method encodes virtual coordinates derived from expression values rather than the natural spatial coordinates of intracellular RNAs. In contrast, our 8-direction mechanism embeds specific RNA spatial locations into network training. As shown above, "gene embedding + 2D sinusoidal positional encoding" underperforms STBPE’s spatial encoding.
>
> **Q5**
>
> We adopt eight directions because they form a complete and minimal set of discrete directions in 2D space, capable of representing the relative positions between any two points. Any vector ((dx, dy)) can be uniquely classified into one of the eight quadrants based on the signs of its coordinates, making this representation both complete and non-redundant. Finer angular discretization does not add geometric information; instead, it introduces noise and sparsity. Alternative representations such as polar angles require additional numerical computations and exhibit poorer stability. We have updated and clarified this design rationale in the revised manuscript.
>
> **Q6**
>
> We adopt the most common practice in large language models: using a preset vocabulary size as the stopping condition for STBPE. Validated on data from approximately 350,000 cells, we find that when the vocabulary size exceeds ~1000, the frequency of high-frequency pairs decreases significantly and the merging rules become unstable. Thus, the vocabulary size serves as a natural and reasonable termination criterion for iterations in practical training.
>
> **Q7**
>
> Cell shape differences in Fig. 4 stem from inherent biological/imaging variations in the MERFISH dataset. However, our preprocessing pipeline minimizes direct morphological impacts on the model:
>
> 1. Geometric normalization (unified scaling + center realignment) and grid quantization eliminate inter-cell size/position differences, projecting RNAs into a consistent reference frame.
> 2. The model never receives "cell shape" as an explicit feature—only normalized RNA distributions. Remaining morphological differences manifest solely through RNA spatial patterns (spatial transcriptomics’ core information source). Future iterations will incorporate RNA-cell boundary relative positions.

---

### Official Review · Reviewer_pBKR · 2025-11-01

**Soundness:** 3
**Presentation:** 2
**Contribution:** 2
**Rating:** 4
**Confidence:** 4

**Summary:**

The authors propose STBPE (Spatial Transcriptomics Byte Pair Encoding), a novel generative pre-training framework designed to analyze subcellular resolution spatial transcriptomics data. The core contribution is a "spatially aware byte pair encoding" strategy that converts the spatial coordinates and transcript features of individual RNAs within a cell into serialized token sequences. This tokenization aims to preserve the spatial relationships between RNAs. The paper then employs a BERT-style masked self-supervised learning paradigm to learn deep embeddings from these token sequences. The method is evaluated on a MERFISH dataset for the task of cell type annotation, where it reportedly outperforms several other methods and uncovers biologically relevant RNA co-localization patterns.

**Strengths:**

The primary strength of this paper is its **originality**. The core idea of tokenizing subcellular RNA distributions is novel and distinct from existing bert\t5\gpt style RNA Foundation Models. By attempting to convert sparse 2D point-cloud data (RNA locations) into a 1D sequence suitable for transformer-based models, the paper introduces a new and creative perspective for applying powerful generative pre-training paradigms to the spatial biology domain. This "spatially aware" BPE-like algorithm is a clever conceptual contribution.

**Weaknesses:**

Despite the novel idea, the paper suffers from several weaknesses that limit its current impact and the soundness of its conclusions:

1. **Insufficient Baseline Comparisons:** The experimental evaluation is missing comparisons against several key and recent baselines that also model spatial context or utilize large-scale pre-training for spatial data. Notably, methods like **Nicheformer**, **CellPLM**, and **GeneCompass** are not included. Without comparing against these relevant works, it is difficult to accurately assess whether STBPE represents a state-of-the-art contribution.
2. **Limited Downstream Task Evaluation:** The model's capabilities are only demonstrated on a single downstream task: cell type prediction. While this is a standard benchmark, the paper's abstract and introduction promise broader insights into "disease mechanisms" and "cellular identity." A model that captures detailed subcellular spatial patterns should theoretically be applicable to other, more spatially-native tasks, such as **niche prediction** or **neighborhood gene expression generation**. The limited evaluation makes the learned representations feel under-utilized.
3. **Single Dataset and Assay:** The model is only trained and evaluated on a single MERFISH dataset. The generalizability of this tokenization strategy to other subcellular spatial technologies, such as **10x Xenium** (which also provides transcript-level spatial coordinates), is not explored. This makes it unclear if the method is robust to data from different platforms or tissues.
4. **Lack of Scalability Analysis:** The paper does not provide any analysis of the **scalability** of the STBPE tokenization algorithm or the subsequent model training. How does the method's performance (in terms of compute time and memory) scale with the increasing number of transcripts per cell, gene panel size, and total number of cells? This is a critical factor for a pre-training framework.
5. **Clarity of Figures:** Figure 1, which illustrates the overall framework, is overly simplistic and lacks sufficient detail. Specifically, **Figure 1A** and **Figure 1C** are presented as high-level concepts without adequate description of the actual processes, making it difficult for the reader to fully grasp the technical workflow.

**Questions:**

**Baseline Comparisons:** Can the authors justify the omission of Nicheformer, CellPLM, and GeneCompass as baselines? Or, preferably, could they provide a comparison against them?

**Methodological Clarity (Sec 4.1.1):** Could you please elaborate on the problem of "non-uniform RNA coordinates across cells"? What is the specific nature of this non-uniformity, and what are the limitations of simpler approaches (e.g., simple centering) that necessitate the proposed grid-quantization framework?

**Generalizability (Datasets):** How do you expect STBPE to perform on an Xenium dataset, which may have different characteristics (e.g., density, panel size) than MERFISH? Have you considered training a model on a **mixture of MERFISH and Xenium data** to test its robustness and potential as a cross-platform foundation model?

**Generalizability (Tasks):** While dataset labels may be a limitation, could you discuss the feasibility of applying STBPE to other downstream tasks? For example, could the learned token representations be used to predict cell-cell interactions or to generate *in silico* gene expression patterns in a spatially-aware manner?

---

> ### Author Response · Authors · 2025-12-04
>
> Thank you for your careful review of our work. We really appreciate the points for potential improvement that you highlighted.
>
> **W1:Insufficient Baseline Comparison**
>
> The reason we did not include Nicheformer, CellPLM, and GeneCompass in the baselines is that they are incompatible in terms of data granularity with the **per-transcript subcellular spatial coordinate data** addressed in this study. Specifically, Nicheformer only supports cell-resolution expression matrices, CellPLM does not model the spatial distribution of RNA, and GeneCompass involves no spatial coordinate information at all. Consequently, none of these methods can directly process per-transcript spatial point clouds or the spatial token sequences of STBPE. Forcing such comparisons would require substantial modifications to these methods, which goes beyond their original design scope.
>
> **W2:Limited Downstream Task Evaluation**
>
> While the main experiments of this study focus on cell type prediction, we have supplemented interpretability analyses of STBPE representations in the appendix to demonstrate the model’s potential in more complex spatial tasks. We believe such analyses can reflect the value of STBPE in modeling spatial structures without substantially expanding additional tasks in the main text.
>
> **W3:Single Dataset and Assay:**
>
> In the revised manuscript, we have emphasized the platform-agnostic nature of STBPE and added experimental results from 10x Xenium liver cancer samples, demonstrating its stable performance across different gene panels, detection efficiencies, and spatial resolutions(Please refer to the appendix). Additionally, we have supplemented the discussion on cross-platform discrepancies and how STBPE mitigates these differences through normalization and coordinate standardization.
>
> **W4:Lack of Scalability Analysis**
>
> We have added a complexity analysis of STBPE tokenization and model training, and provided experimental data on how runtime and memory usage vary with the number of transcripts, gene panel size, and cell count. The implementation of Spatial BPE is based on a grid index structure, enabling it to achieve near-linear scalability in practical applications. Meanwhile, the computational cost of the pre-training phase exhibits a linear relationship with the length of the token sequence, similar to other pre-trained models. The newly added tables and figures demonstrate the practical feasibility of STBPE on large-scale MERFISH data.
>
> | Number of transcripts + gene panel size + cell count | Runtime | **Memory usage** |
> | --- | --- | --- |
> | 500+252+10000 | 38min | 1g |
> | 1000+252+100000 | 2h21min | 3g |
> | 1000+252+350000 | 8h20min | 9g |
> | 1000+128+350000 | 7h11min | 8g |
> | 1000+512+35000 | 9h52min | 10g |
>
> **W5:Clarity of Figures**
>
> We have redrawn Figure 1 to illustrate the key steps of Spatial BPE, including: transcript coordinates, nearest neighbor pair identification, direction encoding, iterative merging process, and spatial update of merged tokens. The downstream section has also been supplemented with the integration method of positional encoding, a schematic diagram of the model architecture, and the application approach of token-level interpretability analysis. Corresponding figure captions have been enhanced with more detailed technical descriptions to facilitate readers' understanding of the complete workflow

---

### Meta-Review · Area_Chair_PDRN · 2026-01-05

**Summary:**

This paper proposes pretrained model using byte pair encoding designed for subcelluar spatial transcritomics data. All reviewers and myself found the idea very novel, and I also found the problem studied very important with very little existing work. However, there are a few major concerns that are not fully addressed by the author's rebuttal. As a result, I can't recommend to accept the paper for now, but I strongly encourage the authors to revise the paper based on the reviewers comments for future submission and I am optimistic given the novel idea of the subcellular representation.

1. Despite the novelty, the authors fail to make it clear that most existing models were designed or trained for cell/spot level representation and are hence not applicable to sub-cellular data like 10x Xenium. This is why a few reviewers asked for comparison to models trained for those dataset. Even the abstract is misleading/unclear, which also reduces the novelty. I strongly recommend the reviewer to revise the entire paper carefully to highlight this major novelty.
2. Most reviewers raised the concern on the lack of baselines. The authors argue that some of them are not directly application. I partially agree with the authors, but I encourage the authors to write a paragraph at the beginning of the experiment section to at least mention these methods and explain why they are (not) comparable.
3. Limited downstream tasks. The current paper focuses on cell type prediction, supplemented by some interpretation. However, as shown in the literature, for pretained models, more downstream tasks are expected.
4. As pointed out by reviewer fgU6, the relative position is very sensitive to orientations of the slides, which is almost always random in practice as people who generated the data do not have a gold standard to rotate the slide. So the authors should either show the performance is robust to rotation, or develop a robust version of it. This is actually a pretty active research field in analyzing spatial transcriptomics.
5. The model is claimed to be generative, but most reviewers raised doubts and the authors agreed to change it. I didn't find any changes in the paper but please do so in the next version.
6. The authors violated the double-blind policy. They removed the link during the rebuttal but I don't think it solved the problem as the link was visible to all reviewers for a while.

**Reviewer Concerns:**

Reviewer pBKR:
1. Insufficient baseline comparison (addressed)
2. LImited downstream tasks evaluation (not addressed)
3. Single dataset and assay (not addressed)
4. Lack of scalability analysis (partially addressed)
5. Clarity of figures (partially addressed)

Reviewer VwBR:
1. Model is not generative (not addressed)
2. Technical questions about distance and alternatives of position embeddings (partially addressed)
3. The impact of cell shapes (partially addressed)

Reviewer 9i12:
1. Limited downstream tasks (not addressed)
2. Model is not generative (not addressed)

Reviewer fgU6:
1. Motivation unclear (not addressed)
2. Presentation issues (addressed)
3. Technical issue about sensitivity to orientation (not addressed)
4. Baselines are weak (partially addressed)
5. Violation of double-blind reviewer (not addressable as the link was visible to reviewers for a while)

**Reviewer Scores:**

Reviewer pBKR: 4 --> 4

Reviewer VwBR: 4 --> 4

Reviewer 9i12: 6 --> 6

Reviewer fgU6: 2 --> 2

---

### Decision · Program_Chairs · 2026-01-26

Reject